# Interpretable classification of Alzheimer's disease pathologies with a convolutional neural network pipeline

Ziqi Tang [1,2], Kangway V. Chuang [1], Charles DeCarli[3], Lee-Way Jin [4], Laurel Beckett [5], Michael J. Keiser [1] & Brittany N. Dugger [6]

Neuropathologists assess vast brain areas to identify diverse and subtly-differentiated morphologies. Standard semi-quantitative scoring approaches, however, are coarse-grained and lack precise neuroanatomic localization. We report a proof-of-concept deep learning pipeline that identifies specific neuropathologies—amyloid plaques and cerebral amyloid angiopathy—in immunohistochemically-stained archival slides. Using automated segmentation of stained objects and a cloud-based interface, we annotate > 70,000 plaque candidates from 43 whole slide images (WSIs) to train and evaluate convolutional neural networks. Networks achieve strong plaque classification on a 10-WSI hold-out set (0.993 and 0.743 areas under the receiver operating characteristic and precision recall curve, respectively). Prediction confidence maps visualize morphology distributions at high resolution. Resulting network-derived amyloid beta (Aβ)-burden scores correlate well with established semi-quantitative scores on a 30-WSI blinded hold-out. Finally, saliency mapping demonstrates that networks learn patterns agreeing with accepted pathologic features. This scalable means to augment a neuropathologist's ability suggests a route to neuropathologic deep phenotyping.

[1] Department of Pharmaceutical Chemistry, Department of Bioengineering and Therapeutic Sciences, Institute for Neurodegenerative Diseases, and Bakar Computational Health Sciences Institute, University of California, San Francisco, 675 Nelson Rising Ln Box 0518, San Francisco, CA 94143, USA. [2] School of Pharmaceutical Sciences, Tsinghua University, 100084 Beijing, China. [3] Department of Neurology, University of California-Davis School of Medicine, 4860 Y Street Suite 3700, Sacramento, CA 95817, USA. [4] Department of Pathology and Laboratory Medicine, University of California-Davis School of Medicine, 2805 50th Street, Sacramento, CA 95817, USA. [5] Department of Public Health Sciences, University of California-Davis, Medical Science, 1C One Shields Avenue, Davis, CA 95616, USA. [6] Department of Pathology and Laboratory Medicine, University of California-Davis School of Medicine, 3400A Research Building III Sacramento, Davis, CA 95817, USA. Correspondence and requests for materials should be addressed to M.J.K. (email: keiser@keiserlab.org) or to B.N.D. (email: bndugger@ucdavis.edu)

Extracellular deposition of amyloid-beta (Aβ) plaques is a pathological hallmark of Alzheimer's disease (AD)[1,2], a common neurodegenerative disease. Aβ plaques have a diverse range of morphologies and neuroanatomic distributions[1]. The current consensus criteria for a neuropathological diagnosis of AD[3–5] incorporate protocols assessing plaque density and distribution; some researchers have hypothesized that plaques may be an initiating event in AD[5,6]. More precise measures of plaque morphologies (such as cored, neuritic, and diffuse) can serve as a basis for understanding disease progression and pathophysiology, providing guidance and insight into disease mechanisms[2,7–10].

For neuropathologic diagnosis, established semi-quantitative scales are used to assess plaque burden (Fig. 1a)[4,8,11,12]. The standard semi-quantitative criteria put forth by the Consortium to Establish a Registry for Alzheimer's Disease (CERAD) based on the manual assessment of the highest density of neocortical neuritic plaques[4,13]. Diffuse plaques, which may be the initial morphological type of Aβ[14,15], can account for over 50% of plaque burden in preclinical cases but are not included in CERAD[16]. Furthermore, data on anatomical location (i.e., Thal amyloid phase) are based on the presence of plaques regardless of type or density[5]. The potential for neuropathologic deep phenotyping efforts that account for anatomic location, diverse sources of proteinopathy, and quantitative pathology densities motivates the development of effective and scalable quantitative methods to differentiate pathological subtypes[17–19].

Existing quantitative methods, such as positive pixel count[20] algorithms, typically rely on human-defined[21] red-green-blue (RGB) or hue-saturation-value (HSV) ranges (i.e., pixel color and intensity) and are thus sensitive to batch differences or to the variable effects of formalin fixation on tinctorial properties. Manual counts or stereological[22,23] methods can be tedious, difficult to score, and time-consuming. Consequently, studies using limited-range scores or overall pathology burden[3,8,13,24] are powerful but have interrater variability,[4,13] are difficult to adapt to statistically meaningful disease-correlation analysis, or are blind to selected locational vulnerability[20]. New methods introducing detailed and sensitive quantification of pathologies would reduce the burden placed on pathologists, increase reliability, and enable studies at a scale that is currently prohibitive.

Deep learning has transformed medical image analysis[25,26]. Convolutional neural networks (CNNs) have achieved expert-level performance in complex visual recognition tasks, including the diagnosis of skin[27] and breast[28,29] cancers. These flexible models learn to recognize intricate patterns directly from visual data without the need for manually-defined image features or expert-delineated templates, and can account for non-trivial variations in image quality and color. In neuropathology, deep learning approaches have been reported for the classification of AD pathophysiology in magnetic resonance and positron-emission tomography images[30–33], and for relating gene expression to neuropathology datasets[34].

We hypothesized deep learning methods could augment neuropathological whole slide image (WSI) analysis[35]. Despite their strong predictive power, deep learning models have been criticized for their poor interpretability and reliance on massive annotated datasets[36]. At the outset, we recognized these factors represented significant challenges in the development of useful tools for neuropathology. An approach tailored to neuropathology would require (1) careful delineation of the machine learning task; (2) construction of a curated image dataset with high-resolution annotations by experts; and (3) extensive model interpretability. As a proof of concept, we posited CNN models could be employed for recognition and classification of Aβ pathologies, especially plaques, with the downstream goal of providing reliable, scalable, and interpretable measures based on neuroanatomical location. To develop a useful tool to aid

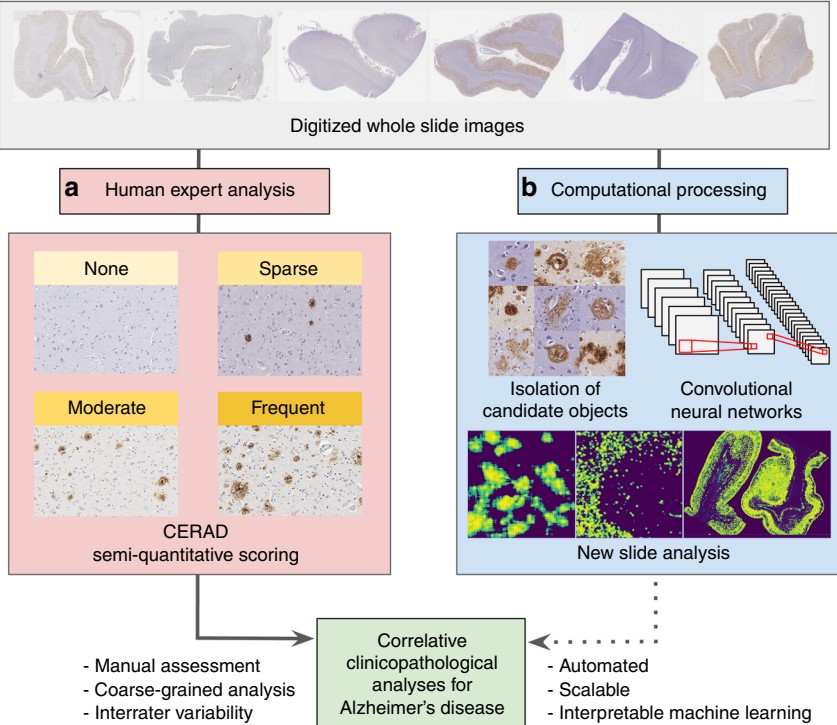

**Fig. 1** Ways of assessing neuropathologies in human tissues. **a** Current protocols for neuropathological assessment of WSIs typically rely on comparatively coarse-grained semi-quantitative scoring such as CERAD[4]. **b** We report an automated computational approach to process entire digitized immunohistochemical stained archival slides, leveraging convolutional neural networks for amyloid plaque classification and localization

neuropathologists, we deemed it critical that predictive performance should result from learning meaningful patterns within the images[37].

In this study, we present a pipeline for the neuropathological analysis of Aβ pathologies in WSIs generated by digitizing glass microscope slides of temporal gyri of the human brain (Fig. 1b). We describe an end-to-end pipeline for image processing, a custom web interface for rapid expert annotation, and training of CNN models that result in high performance multi-task classifiers capable of distinguishing Aβ pathologies in the form of cored plaques, diffuse plaques, and cerebral amyloid angiopathy (CAA). We demonstrate how prediction confidence maps visualize distributions as an interpretable and complementary means to understand Aβ burden. Finally, we provide visual evidence that these models are interpretable, using deep learning introspection methods to show that trained models learn relevant features of each of these Aβ pathology classes. To the best of our knowledge, these studies constitute the first report of CNNs for Aβ pathology analysis.

## Results

**A platform to rapidly annotate 77,000 plaque candidates.** CNNs operate most effectively when trained on datasets exceeding tens of thousands of example images[38]. Indeed, we found that 43 digitized glass microscope slides (WSIs, see Supplementary Table 1 for case details) used in this study yielded over 500,000 individual candidate objects of interest. We set out to build a dataset of approximately 50,000 annotated images for model training (Fig. 2, see Data Availability). Manual annotation at this scale would have been a daunting task through conventional, hand-drawn bounding boxes on a standard ~700 micron visual field. Using open-source image analysis tools (see Methods), we developed an automated preprocessing procedure (Fig. 2a and Supplementary Figs. 1, 2) to normalize slide color and generate bounding boxes around all immunohistochemically (IHC)-stained objects within WSIs. As the native resolution of a WSI is too large (typically 50,000 by 50,000 pixels at ×20 magnification) to use as the direct input for CNNs, we designed the dataset to contain uniform $256 \times 256$ pixel tiles centered on individual plaque candidates.

We created a simple web interface to rapidly annotate Aβ pathology-candidate image tiles and deployed it on the Amazon Web Services Elastic Beanstalk[39] for reliability and scalability (illustrated in Fig. 2c). An expert neuropathologist annotator used unique credentials and a rapid keystroke-entry format to annotate the tiles, which were stored in a standardized query language (SQL) database (see Supplementary Fig. 3). Using this platform, candidate images were annotated at rates up to 2500 tiles per hour into three major categories—cored plaques, diffuse plaques, or CAA. Additional categories such as not sure or flag denoted uncertainty, image segmentation failures, or other special cases (Supplementary Fig. 3). The dataset was then built in three phases (Table 1). In Phase I, 55,001 images were expert-labeled using the web application. The majority of candidate images were annotated as diffuse plaque morphologies (84.8% of the annotations), with cored plaques (2.2%) and CAAs (1.1%) making up the minor classes (Table 2). Furthermore, of the CAA annotated images, a second annotation step divided the group into capillary (36.3%) and non-capillary (63.7%). As class balance typically improves machine learning model performance, we sought to enrich the minority classes in a second phase. We trained an intermediate CNN to classify objects based on the Phase I dataset, then used its predictions to prioritize an additional set of 101,671 unprocessed tiles in favor of cored plaques and CAAs for manual annotation (see Supplementary

Fig. 4). Thus in Phase II, an additional 11,029 tiles were annotated, having been evaluated in rank-order of their predicted likelihood to contain either of the minority-class plaques. In Phase III, we annotated an additional 10,873 candidate tiles extracted from a separate hold-out test set of 10 WSIs not in the original 33-WSI collection, without any prioritization procedures. We performed manual annotation using the web application for all phases.

**CNNs effectively discriminate among Aβ morphologies.** We trained CNNs to classify tiles as containing cored plaque, diffuse plaque, and/or CAA. At ×20 magnification, a single $256 \times 256$ pixel tile (128 microns) could contain more than one object, so we trained a multi-task CNNs for multi-label classification: CNNs were asked to determine the presence or absence of all morphologies in each tile. We combined the Phase I and Phase II datasets, then randomly split the resulting 70,000 tiles (66,030 annotated and 3970 IHC-negative) into training (from 29 WSIs) and validation (from 4 WSIs) sets, while stratifying by case (i.e., WSI source) to ensure that models generalize to new cases. A search of CNN architectures identified a six-layer convolutional architecture with two dense layers (Fig. 3a) with strong performance. Using subsequent hyperparameter optimization we found data augmentation[40,41] and minority class oversampling[42] training procedures (Supplementary Fig. 5, Methods) yielded a pronounced performance boost. For completeness, we also recapitulated the analyses without WSI color normalization, but saw no substantive change in performance (Supplementary Fig. 6).

The resulting CNN model trained on 61,370 example tiles achieved validation set performance of 0.983 area under the receiver operator characteristic (AUROC) (Supplementary Fig. 7a) with an area under the precision-recall curve (AUPRC) of 0.845 (Supplementary Fig. 7b). On the strict hold-out (Phase III) test set, the model likewise generalized well to unseen decedent cases (AUROC = 0.993, AUPRC = 0.743, Fig. 4a, b). CAA prediction performance was also strong on the validation set (Supplementary Fig. 7), but was omitted from the Phase-III test set benchmarking in Fig. 4 because Phase III was derived from cases that were predominantly lacking in CAA in those specific regions (e.g., Supplementary Fig. 8). The overall classification accuracy was 0.973 on the validation set and 0.987 on the hold-out test set (Supplementary Tables 3–5). Notably, model performance was achieved using fewer than 2000 training examples of each minority class (cored plaques and CAAs). Representative accurate (Fig. 3b) and misclassified (Fig. 3c) examples from the 10,873 hold-out set tests illustrate cases where the model succeeded or went astray.

**Performance improves nonlinearly with training example count.** To determine whether similar performance could be achieved with fewer manual annotations, we performed two retrospective studies to investigate the effect of training dataset size. In the first study, we randomly selected subsets of the 61,370-example training dataset, maintaining stratification by case (i.e., WSI source), and plotted model performance as a function of the number of training examples (Fig. 4c). Each random selection was repeated five times, and a fresh model trained each time, for a total of 90 independently trained and evaluated CNN models with identical architectures. All models were benchmarked against the same hold-out (Phase III, as in Fig. 4a, b) test set. As expected, model performance positively tracked with the total number of training examples. Notably, models trained on a 50% smaller training set size still achieved an

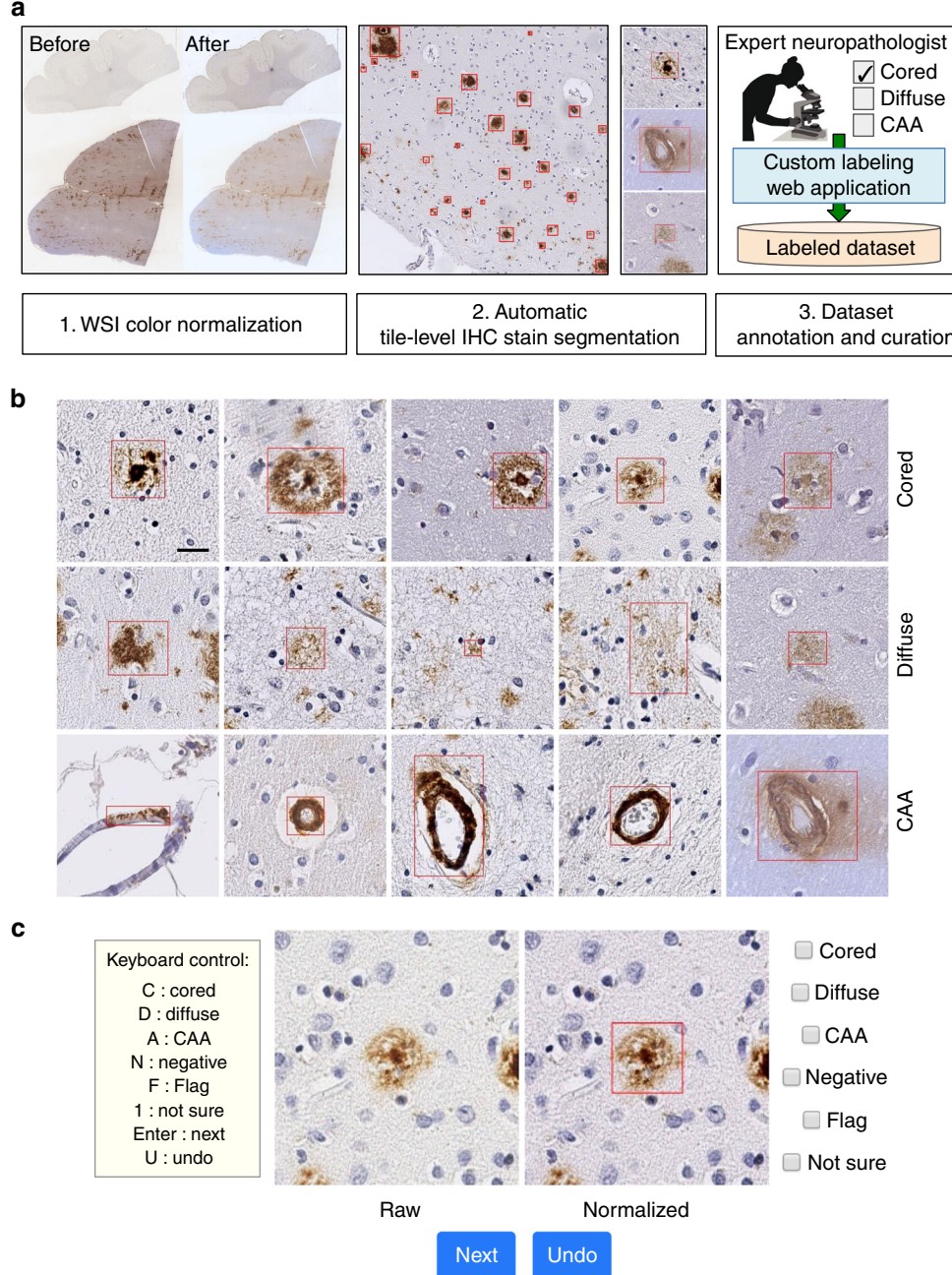

**Fig. 2** Creation of an annotated plaque and CAA dataset for machine learning. **a** Summary of the image processing pipeline, including color normalization, IHC stain segmentation, and extraction of candidate objects (red boxes), followed by rapid expert annotation using a cloud-based web application. **b** Examples of extracted cored plaques (top row), diffuse plaques (middle), and CAA (bottom) and their surrounding tissue area. Rectangles shown in red bound the candidate object during the labeling process. Scale bar = 25 μm. **c** A custom web interface allows for the rapid annotation of plaques by mouse or keystroke, with visualization of raw (without color adjustments) and normalized images, showing the object bounding boxes around which the tile is automatically centered and cropped

| Table 1 Summary of annotated object tile dataset by project phase | | | | |
|---|---|---|---|---|
| **Phase** | **Cored plaque** | **Diffuse plaque** | **CAA** | **Total** |
| Development phase I | 1233 (2.24%) | 46,650 (84.82%) | 778 (1.14%) | 55,001 |
| Development phase II | 1035 (9.38%) | 7610 (69.00%) | 1405 (12.74%) | 11,029 |
| Development total | 2268 (3.43%) | 54,260 (82.17%) | 2183 (3.31%) | 66,030 |
| Test (phase III) | 83 (0.76%) | 10,234 (94.12%) | 7 (0.06%) | 10,873 |
| Remaining unlisted percentages correspond to Not Sure or Flagged labels (see Methods) | | | | |

| | Images | Cored plaque | Diffuse plaque | CAA | AUROC (cored) | AUPRC (cored) |
|---|---|---|---|---|---|---|
| **Table 2 Annotated object dataset distribution by class and model performance** | | | | | | |
| Train | 61,370 | 2141 | 48,123 | 2227 | 0.993 | 0.981 |
| Validation | 8630 | 381 | 7487 | 126 | 0.983 | 0.845 |
| Test | 10,873 | 98 | 10,480 | 7 | 0.993 | 0.743 |

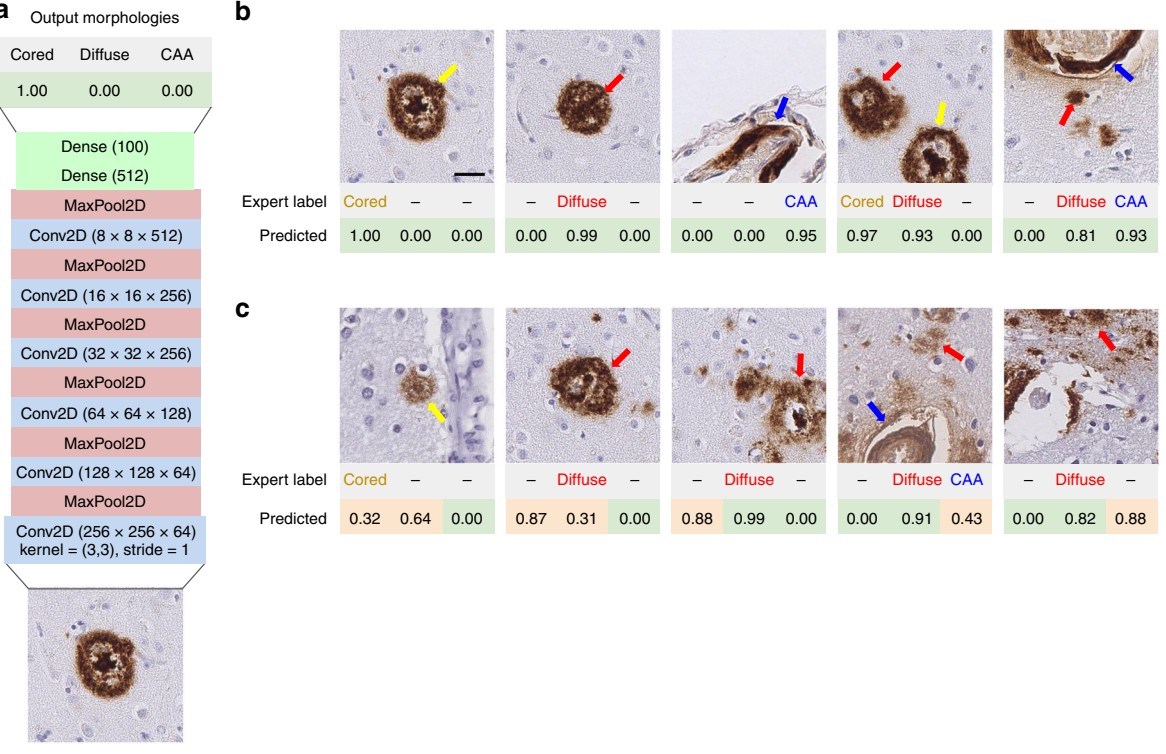

**Fig. 3** CNN models identify three Aβ deposit types in image tiles. **a** The optimized CNN model architecture contained six convolutional layers and two dense layers, using exclusively 3 × 3 kernels and alternating max-pooling layers. **b** Examples of correct CNN predictions. The ground truth expert label row indicates the pathologies that had been manually found within the tile image. The predicted row shows corresponding model confidences for cored plaque (yellow arrow), diffuse plaque (red), and CAA (blue) classes (from left to right). Model predictions range from 0.00 to 1.00, where a higher score indicates higher predicted confidence by the CNN for that plaque class (e.g., the 1.00 corresponds to 100% model confidence that a cored plaque is present in the leftmost panel). **c** Examples of CNN predictions that do not agree with the expert manual annotation. Incorrect model predictions are indicated by light orange backgrounds in the predicted column; green backgrounds correspond to correct predictions. Scale bar = 25 μm for all images

average AUROC above 0.99 and an AUPRC above 0.74, at minimal loss to overall performance.

In the second study, we investigated model performance as a function of the chronological dataset growth during the project, where training examples were included in the order of original expert annotation (Fig. 4d). Model performance at 15 expert-hours fell short of model performance at 50% of dataset size (Fig. 4c). Accordingly, the goal of this second study was to determine whether annotation chronology played a role in CNN training. As above, performance steadily increases as the annotated dataset grows. However, performance trends between the studies differed in two ways. Chronologically-trained models did not converge in AUPRC performance as early as the equivalent-sized random-subset-trained models benefitting from later annotations did. Second, the chronology study shows a distinct AUPRC boost in Phase II, illustrating the positive effect of enriching for cored-plaque prevalence.

**Prediction confidence maps show plaque localization.** To visualize the distribution and neuroanatomic location of Aβ pathologies in a broader context, we applied a sliding window approach[43] to generate WSI heatmaps of predictions (Fig. 5). These heatmaps plot the confidence and location of each prediction by the CNN, which may then be visualized from the sub-tile resolution (Fig. 5c) up to the full WSI view (Fig. 5a). By progressively zooming in from larger anatomical views, the visualization shifts from the broad distribution of plaques to their detailed ×20 morphology. A single cored plaque can be distinguished from a dense region of neighboring diffuse plaques (Fig. 5c). In this cohort, diffuse plaques are densely distributed across the gray matter, whereas cored plaques are predominantly located in deeper and lower cortical layers, in accordance with known neuroanatomic distributions[1]. Furthermore, CAA predictions predominantly appear proximal to the cortical surface where leptomeninges are present[44], although predictions are made independently of the surrounding field or broader neuroanatomic context. These maps highlight other locational aspects of the plaques, such as their presence in the white matter immediately beneath the gray matter[45].

**Classification performance does not vary by tissue landmark.** The CNNs perform classification (e.g., Fig. 4) directly on small

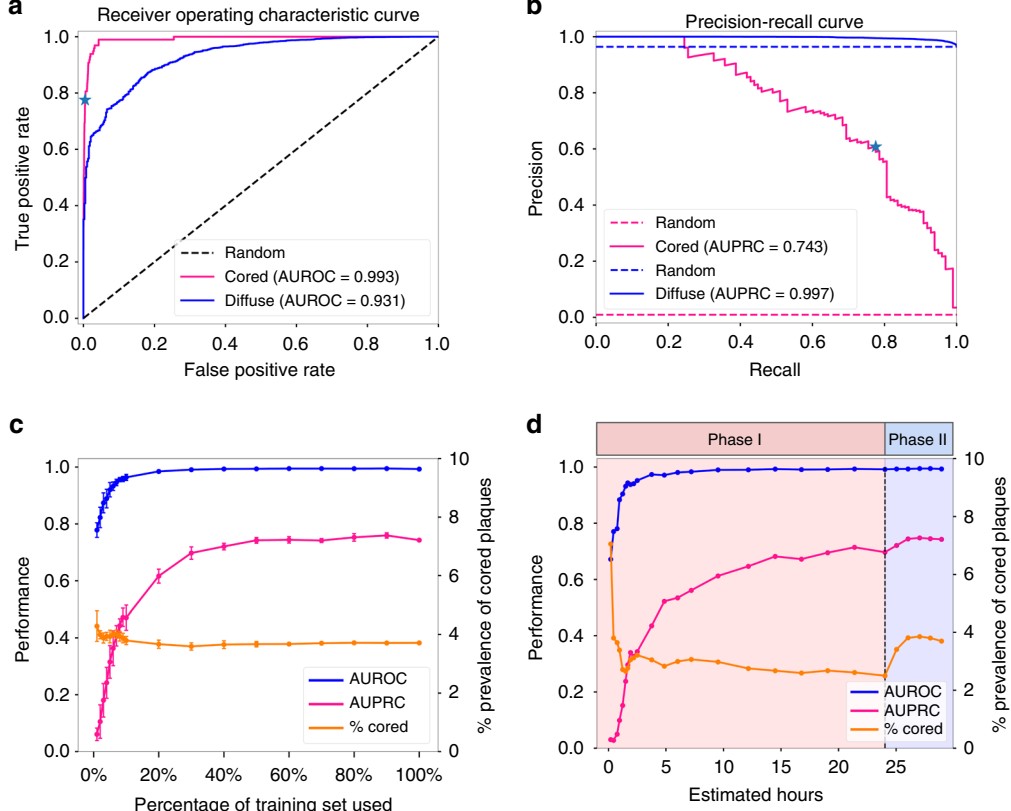

**Fig. 4** Predictive performance on the held-out Phase-II test set ($n = 10,873$). **a** Receiver operator characteristic (ROC) and **b** Precision-recall curves (PRC) for cored (magenta lines) and diffuse (blue lines) plaques. The blue star marks the best trade-off point where prediction confidence threshold equals 0.91. **c** Summarized areas under the ROC and PRC (AUPRC and AUROC) of independently-trained CNNs ($n = 5$ per point) for the task of cored plaque classification, as a function of training dataset size. Dataset was randomly subsetted at each point independent of the date of tile annotation. Error bars represent s.d. **d** AUPRC and AUROC of CNNs for the same task of cored-plaque classification, as a function of chronological dataset growth by annotation timestamp, over the course of the project, showing chronology-dependent dataset effects. Source data are provided as a Source Data file

anatomical areas (128 microns; green box in Fig. 6a). Human experts typically assess larger fields of view such as ~700 microns viewed at ×10 magnification when conducting semi-quantitative plaque scoring. To visualize prediction performance in this context, we also assessed cored-plaque agreement maps on contiguous 6-by-6 tile (768 micron) regions (Fig. 6a). In the leftmost column, a green box surrounds the cored plaque within the tile, as labeled by a neuropathologist during the Phase-III dataset annotation (Fig. 6a). The middle column overlays the prediction map (as in Fig. 5c) onto the original IHC-stained image. Finally, the rightmost column summarizes agreement between the expert label and the prediction, with blue and cyan representing correct prediction areas, while red and orange denote misclassification[46]. For this analysis, we used a CNN prediction confidence threshold of 0.90. A more permissive threshold would decrease false negatives (red) at the cost of more false positives (orange). Interestingly, this agreement-map highlights the limitations of bounding-box annotations, such that the correct cored-plaque prediction shown is nonetheless penalized by this view (red halo) for accurately predicting the rounded boundaries of the actual plaque instead of anticipating its square ground truth bounding-box.

Stepping further out to regions of 3840 microns (Fig. 6b), these maps (see Supplementary Fig. 9 for additional examples) visualize the results plotted in Fig. 4, with the complementary addition of tissue landmarks, prediction clustering, and neuroanatomic localization. The model reliably rejects background tissue and diffuse plaque deposits, while accurately identifying most cored

plaques. Model performance does not change based on the nearby neuroanatomic architecture in these examples, although occasional clusters of co-localized false-positive (orange) cored plaque predictions can appear (e.g., Supplementary Fig. 9).

**Introspection studies identify salient plaque features.** To investigate the CNN model's internal logic, we performed two studies to determine the importance of morphology features contributing to accurate predictions (Fig. 7). In the first, we applied guided gradient-weighted class activation mapping (Guided Grad-CAM)[47] to identify salient visual features underlying the model's predictions. Guided Grad-CAM follows the CNN's gradient flow from individual tasks back onto the original image tile to establish an activation map, highlighting the input features most relevant to each CNN prediction. Figure 7 shows Guided Grad-CAM (white features on black background) on examples taken from cored plaque, diffuse plaque, and CAA classes. Consistent with human expertise, Guided Grad-CAM activation maps predominantly highlight regions of the tiles corresponding to IHC-stained Aβ pathologies. For instance, in Fig. 7a, the activation map for cored-plaque prediction highlights a dense and compact Aβ center (yellow arrow), whereas for the diffuse task the activation map highlights the off-center diffuse object (red arrow). By contrast, the CAA activation map highlights the periphery of the image, much as CAA often forms a ring within vessels, although none could be found. In Fig. 7b, the diffuse-task Guided Grad-CAM

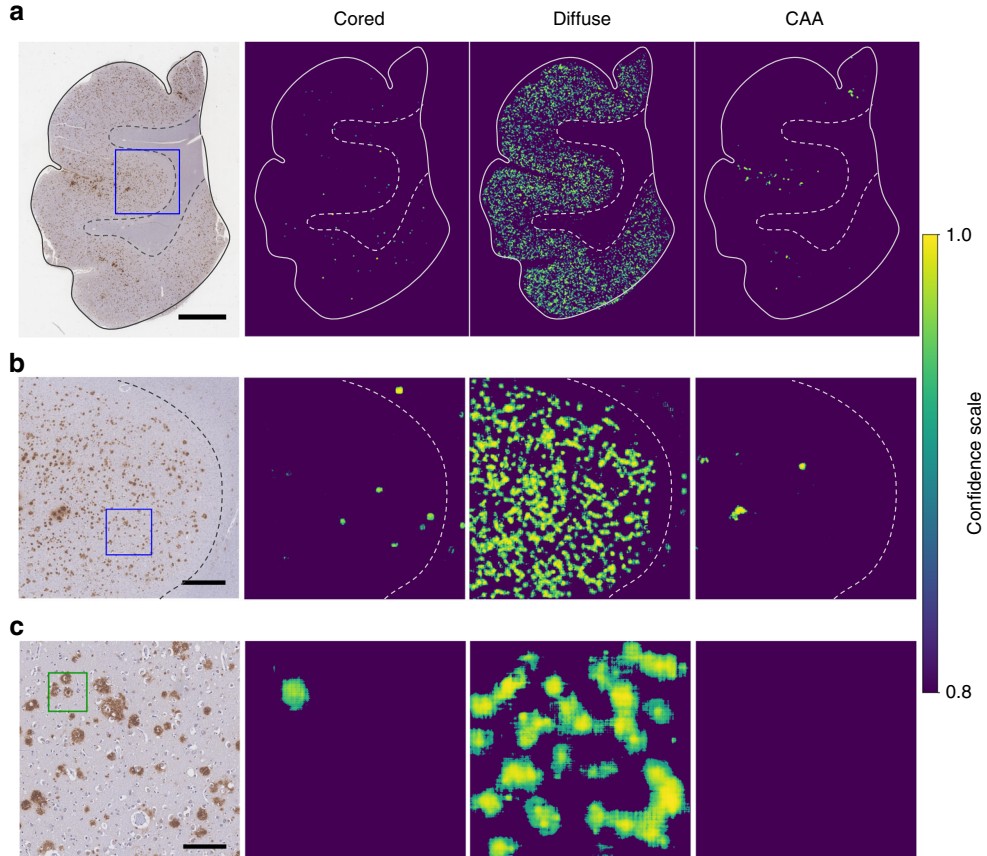

**Fig. 5** Prediction confidence heatmaps for cored plaques, diffuse plaques, and CAA. **a** Whole slide overview visualization, revealing broad amyloid distribution patterns. Scale bar = 3 mm. **b** Higher magnification (×4) view of the blue-boxed region from panel **a**. Scale bar = 750 μm. **c** Higher magnification (×20) view of the blue-boxed region from panel **b**. Green box marks cored plaque manual annotation. Scale bar = 150 μm. Confidence scales for each panel were bottom-capped to aid visualization, such that only confidence scores ≥ 0.8 are plotted, with yellow being the most confidence, and purple the least. Approximate (hand-drawn) boundaries of gray versus white matter (dotted line) and tissue boundaries (solid lines) are overlaid for reference

highlights ill-defined amorphous Aβ pathologies, while for the cored and CAA tasks it focuses on punctate IHC staining and potential microglia. The CAA activation map in Fig. 7c highlights ring structures (blue arrow) within the media of the cortical vessel, consistent with CAA's defining feature; while for cored and diffuse tasks, Guided Grad-CAM highlights the punctate deposit (red arrow) beneath the CAA. Lastly Fig. 7d, which contains both a diffuse (red arrow) and a cored plaque (yellow arrow), shows cored-task activation maps localizing to the amyloid core, with broader feature activations for diffuse and CAA tasks. Crucially, Guided Grad-CAM activation mapping may highlight certain image features as salient because they help determine that an object is not present in the image: Despite strong localized activation for cored and diffuse maps in Fig. 7c at the punctate deposit (red arrow), the CNN predicts that neither plaque is present.

Whereas Guided Grad-CAM provides a fine-grained view of feature salience, it does not differentiate features indicative of a plaque from those that contradict its presence. To complement the analysis, we performed a feature occlusion study[48] on the same examples. In this experiment, a small occlusion patch (shown in Fig. 7a, black box) is systematically moved across the image, and the model makes a prediction on the occluded image at each increment. Blue-to-yellow-to-red colors indicate increasing CNN prediction confidence from 0.0 to 1.0. Consequently, color shifts in occlusion maps show which image features, when

occluded, change prediction confidence. When the patch occludes an important feature such as the amyloid core of a cored plaque (Fig. 7a, yellow arrow), the model fails to predict the object correctly: cored-task confidence drops to zero (blue dot on red background, yellow arrow). Occluding less cored-task-relevant regions such as within the off-center diffuse stain (red arrow) have little effect, indicated by the solid red coloring in the cored-task's confidence map for this area. Conversely, confidence maps may also show where occlusion of a critical feature makes an alternative class more likely. If the amyloid core in Fig. 7a is occluded, diffuse plaque prediction becomes likely (signified by yellow arrow).

Where more than one plaque occur within the same tile, the two feature-importance studies differ. Guided Grad-CAM activation maps identify salient pixels for plaque classes independently, whereas occlusion maps highlight the interplay of features among classes. For example, in occlusion maps, occluding the leftmost plaque decreases diffuse-task confidence (Fig. 7d, light blue region in the diffuse-task map, red arrow), whereas the same prediction gains confidence (red region, yellow arrow) when the cored plaque's core is occluded. In the corresponding Guided Grad-CAM activation maps for the diffuse-task, however, features specific to the diffuse plaque are predominant. Together, these complementary maps visualize the features within images that motivate the CNN's plaque predictions.

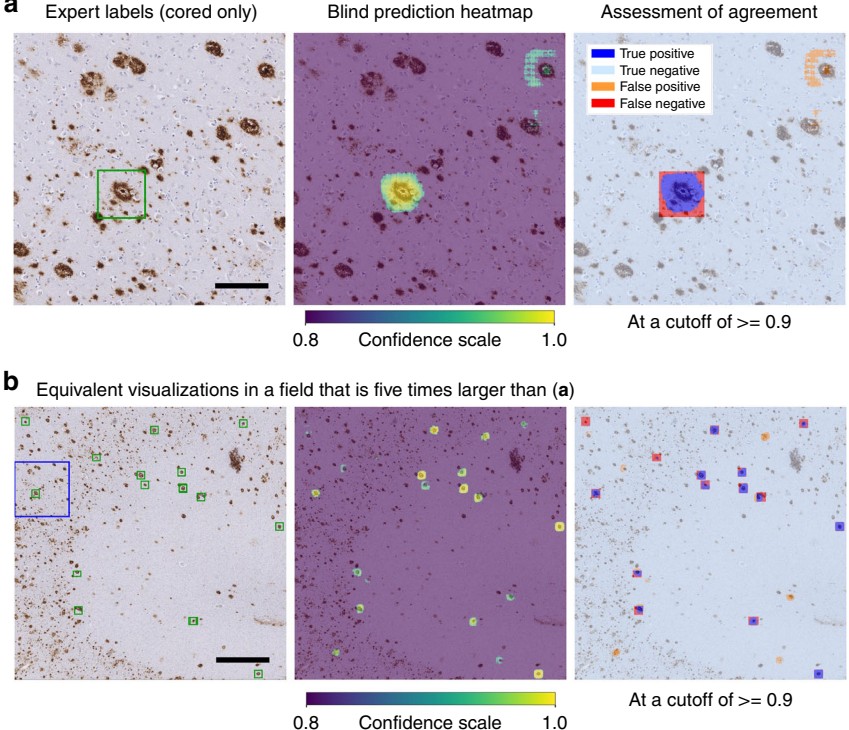

**Fig. 6** Visualization of a representative example from the cored-plaque classification tests plotted in Fig. 4. **a** CNN model prediction confidence maps (middle panel, as in Fig. 5c) overlaid onto the original slide tile. Bounding boxes mark cored-plaque expert annotations (left panel, green box). The combined map (right panel) assesses agreement between the model's predictions and the expert labels, where pixels are colored by a semi-transparent overlay as true positive (blue), false positive (orange), true negative (cyan), and false negative (red) areas. Scale bar = 150 μm. **b** Prediction-versus-annotation agreement map generated as in **a**, but with a larger field for greater tissue and plaque clustering context. Scale bar = 750 μm. Confidence scales for middle panel are bottom-capped to aid visualization, such that only confidence scores ≥ 0.8 are plotted, with yellow being the most confident and purple the least

**CNN-based WSI scores correlate with semi-quantitative scores**. To compare with manual semi-quantitative approaches such as Consortium to Establish a Registry for Alzheimer's Disease (CERAD), we developed a preliminary neural-network derived score for Aβ pathologies at a global WSI level. For the CNN-based score, we calculated a count of each predicted Aβ pathology across an entire WSI by segmenting its prediction heatmap (e.g., Fig. 5a) and normalizing the result by the tissue area of each slide (Supplementary Fig. 10). The resulting CNN-based scores correlated strongly across the total dataset of 62 WSIs (Supplementary Tables 1 and 2) for which we had independently-collected semi-quantitative, CERAD-like scores for each specific class on 62 of the WSIs (Fig. 8 and Supplementary Table 6). CNN-based scores for Aβ pathologies significantly differentiated WSIs by CERAD-like categories (e.g., moderate versus frequent), especially for cored plaques (Fig. 8a, second row). For instance, CNN-based WSI scores between none versus frequent CERAD-like categories were exponentially separated. To better assess generalization, we collected a further set of 20 WSIs (Supplementary Table 2) with corresponding CERAD scores that were blinded during analysis. Combined with the 10 separate hold-out WSIs from Phase III, we found this 30-WSI blinded hold-out set demonstrated strong correlation between the automated and manual scoring approaches, such that CNN-based scores significantly discriminated existing semi-quantitative categories (Fig. 8b).

## Discussion

We report a scalable, quantitative, and interpretable approach to identify neuropathologies for three classes of Aβ pathologies,

motivated by the method's downstream application to statistically powerful correlative analyses and neuroanatomical localization of AD pathologies. In practice, such deep-phenotyping techniques will have limited utility if their underlying predictions cannot be interpreted, critiqued, and refined by expert neuropathologist supervision. Consequently, to establish the feasibility and limitations of this approach, we considered multiple challenges when adapting CNNs to WSIs of archival human brain samples[49]: (1) WSIs are prohibitively large and can vary in color, necessitating a preprocessing pipeline for their normalization, segmentation, and efficient annotation; (2) deep neural networks are notoriously data-hungry; we evaluated how much training data a CNN requires to discriminate Aβ morphologies (cored plaque, diffuse plaque, and CAA); and (3) it is rarely obvious how generalizable a model will be across diverse individuals, populations, and conditions—to prove an enabling tool, model predictions must be interpretable and derive from meaningful features amenable to a neuropathologist's critique and feedback.

Addressing the first challenge, we developed an end-to-end pipeline to automate WSI processing and aid rapid image annotation (Fig. 2a, Supplementary Fig. 3). This pipeline performs color normalization[50] followed by IHC stain detection to create a preliminary library of candidate plaques at ×20 magnification. The logic behind stain detection was two-part: Stained objects inhabit a delimited brown hue range and objects comprise coherent contiguous regions exceeding a minimum size. We then generated image tiles centered on each candidate (Fig. 2b). Thus, 43 WSIs containing Aβ IHC-stained temporal gyri yielded nearly 500,000 raw candidate tiles at ×20 resolution (details in

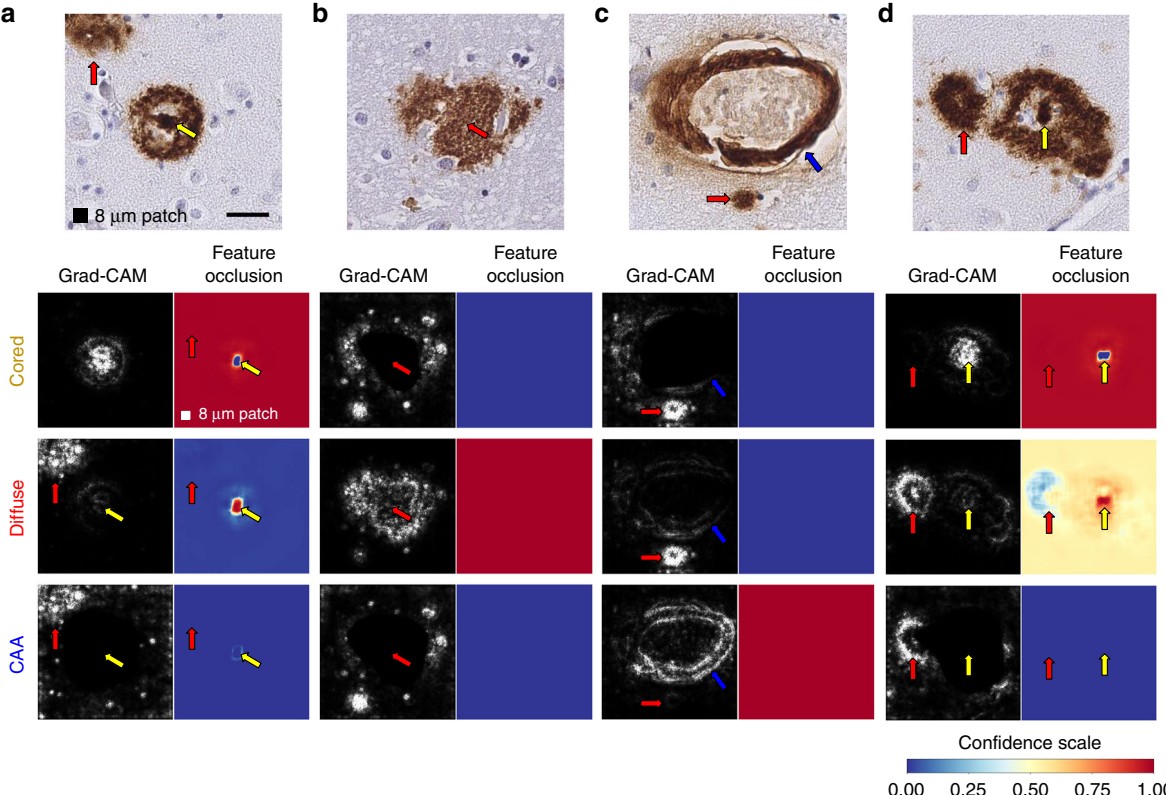

**Fig. 7** Model interpretability studies using machine-learning introspection techniques. **a** A cored plaque example (top row, yellow arrow). For the task of cored-plaque prediction, the activation map (by Guided Grad-CAM; left, second row) and the feature occlusion map (right, second row) identify the amyloid core (yellow arrow) as the defining morphological feature. By contrast, the diffuse stained region (red arrow) only arises as a salient feature during diffuse-plaque and CAA prediction tasks (third and fourth rows, respectively). **b** Diffuse plaque example where activation and feature occlusion maps focus on ill-defined amorphous amyloid contours for diffuse-plaque classification task (third row). **c** CAA example, where the CAA task's activation and feature occlusion maps (fourth row) highlight amyloid ring pixels within the media of the cortical vessel (blue arrow), while for cored and diffuse tasks the small punctate IHC staining is considered salient (red arrow; second and third rows). **d** Example containing both diffuse (red arrow) and cored (yellow arrow) plaques in the same tile illustrate the difference between activation and feature occlusion maps. Confidence scales for feature occlusion maps represent the CNN's prediction confidence on the occluded image, with red being the most confident and blue the least. Scale bar = 25 μm

Supplementary Table 1), filtered to approximately 200,000 tiles containing candidate objects of sufficient size (see Methods). The next step was image annotation for supervised machine learning. Although web-based histopathological annotation tools exist[51,52], we developed a simple platform using the cloud-based Amazon Web Services Elastic Beanstalk[39] infrastructure (Supplementary Fig. 3) for study design flexibility and for the speed of its keystroke-based entry format. For instance, subsequent studies may investigate a broader field of view for annotation context or introduce checks on intra-rater reliability by re-presenting tiles to annotators in different orientations. Given the scope of the annotation task, we incorporated several aspects of gamification theory[53,54], such as annotator leveling, achievement badges, and progress-bar filling[55,56] to acknowledge and motivate progress. Using this tool, we observed sustained annotation rates reaching 1.44 s per tile.

The second challenge was in determining the necessary training dataset size. Having manually annotated 66,030 candidate tiles from 33 WSIs in two annotation phases (Table 1), plus 3970 randomly selected IHC-negative tiles, we examined the CNN's ability to precisely discriminate plaque and CAA morphologies. For this analysis, we randomly split the tiles into train and validation sets, such that train and validation tiles never shared the same WSI source. In addition, as a strict hold-out test set (Phase III) and to investigate the role of nearby neuroanatomic landmarks on prediction, we annotated larger contiguous tissue

regions corresponding to 5× the standard dimensions, for 10 previously unseen WSIs (Supplementary Fig. 8). Note this Phase III dataset differed from the Phase I + II train and validation datasets in that the latter's 70,000 labeled tiles were randomly selected; there was no guarantee that tiles and their resulting expert plaque annotations would be contiguous or comprehensively labeled for any local region of tissue. We trained multi-task CNNs on all 61,370 training tiles, evaluated multiple CNN architectures and hyperparameter choices, and found that a relatively simple model design (Fig. 3a) inspired by Simonyan and Zisserman's VGG[57] achieved strong classification performance (Fig. 4a, b).

Given the substantial time investment, we asked whether similar performance could have been achieved with fewer training tiles. We evaluated this retrospectively, by progressively decreasing the training dataset size in two different ways (Fig. 4c, d). In the first study, we selected a progression of training data subsets randomly and repeated the training process five times per subset size (Fig. 4c). In the second study, we maintained the chronology of the project instead, and plot a natural history of the annotation process. Intriguingly, these performance evaluations highlighted two annotation regimes (Fig. 4d); first, unbiased random-tile candidate labeling (Phase I), followed by the Phase II procedure, where cored-plaque and CAA candidates were purposefully enriched by bootstrapping from a Phase-I-trained CNN model. As expected, increasing training example counts improved

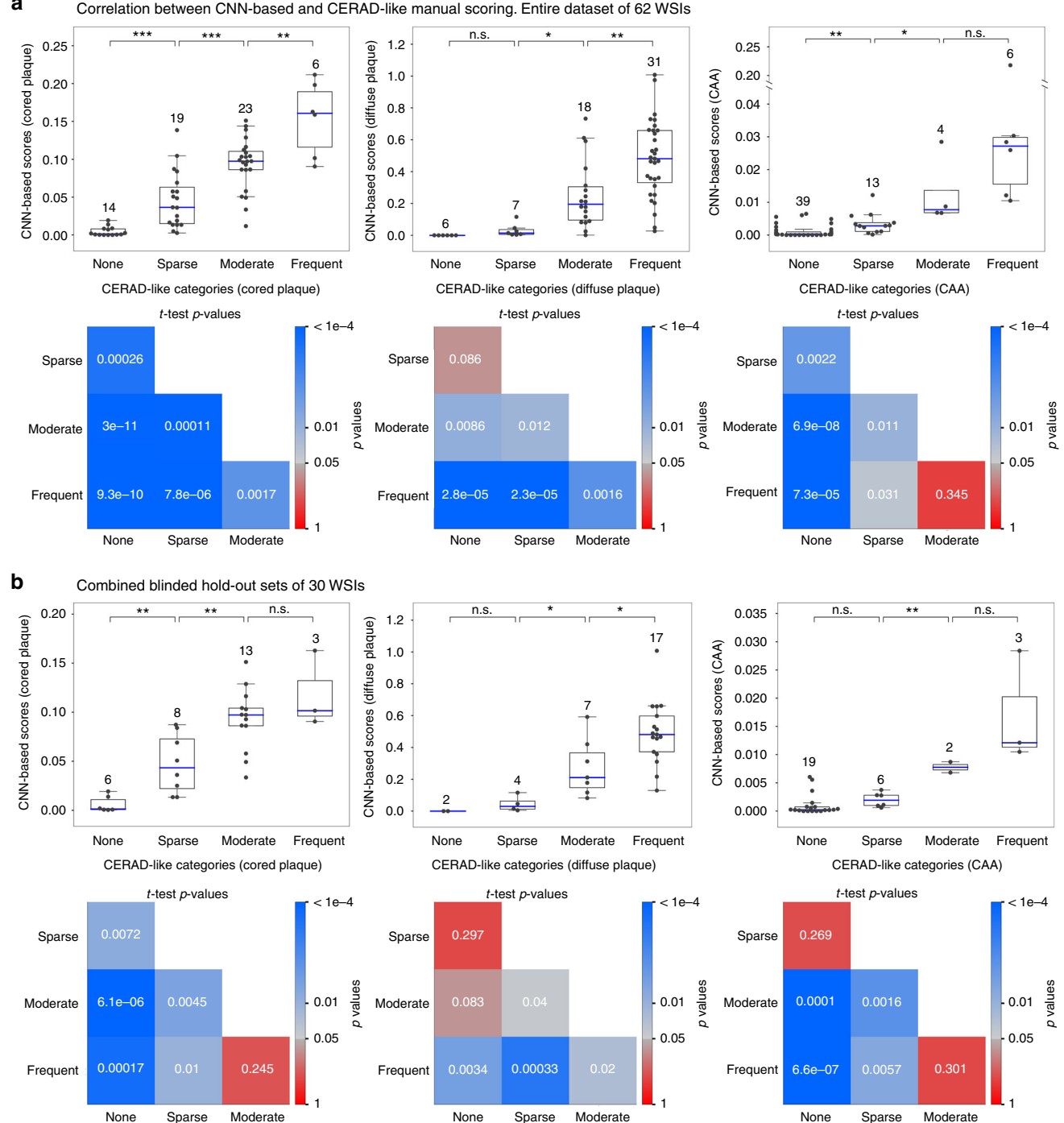

**Fig. 8** Comparison of CNN-based Aβ-burden scores versus manual CERAD-like semi-quantitative scores at a whole-slide level for each pathology. **a** The automatic and manual scores correlate well across the entire dataset of 62 independent WSIs, comprising the original Phase I-III slide set plus 20 additional blinded WSIs. **b** Correlations assessed on the 20 blinded WSIs not used in any previous step of the study combined with the 10 WSIs from the original hold-out set, for a total of $n = 30$ individual cases. Box plots show median (center line inside the box), interquartile range (IQR, bounds of box), minimums and maximums within 1.5 times the IQR (whiskers), and outliers (points beyond the whiskers), with a dot per WSI. $p \geq 0.05$ was considered not significant (ns); *$p < 0.05$, **$p < 0.01$, ***$p < 0.001$. Matrices in the second row of each panel exhaustively plot $p$-values of CNN-based score distributions between all pairs of CERAD-like categories for the corresponding box plot, where squares are colored in log scale by $p < 1e-4$ (blue) to $p = 0.05$ (gray) to $p = 1$ (red; insignificant) using two-sided Student's $t$-tests. Source data are provided as a Source Data file

model performance. Less anticipated was that chronologically-early annotations appeared to be less effective for model training (Fig. 4d); considerations such as the neuropathologist's growing familiarity with the annotation tool and its visual field may be subjects for further study. From a practical perspective, the

steepest performance gains were achieved within the first 15 h of expert labeling, suggesting a reduced dataset may be pragmatically sufficient for classification of cored and diffuse plaques. Significantly, models trained using a comparatively small investment of a neuropathologist's time can assist with new cases and

potentially reduce overall expert burden. Subsequent refinements to the model, particularly in reinforcement feedback on incorrectly-classified examples encountered during the model's use (e.g., Fig. 3d), might later be incorporated into the workflow with minimal friction.

The third challenge was human interpretability. We posited that visualizing the CNN model's predictions as comprehensive confidence maps from the whole-slide level down to a focused plaque-level field (20×) would aid interpretability by a trained neuropathologist, given the importance of local tissue and neuroanatomic context. On a neuroanatomic level, most predicted plaques are located within gray matter (Fig. 5a, yellow-to-green regions, right three columns) with some sparse densities in the white matter not appreciated from the raw slide (Fig. 5a, left column). Despite their primary localization within gray matter, studies have reported plaques within white matter[1,45,58]. Furthermore, the maps predict cored plaques' propensity for deeper and lower cortical layers, consistent with their known neuroanatomic distribution[1,59]. We were likewise gratified to observe that individual cored plaques stand out from dense neighborhoods of diffuse plaques (Fig. 5c, cored column) and that CAA predictions made by the model on a 20 × (128 microns) tile-by-tile basis nevertheless localized predominantly to the leptomeninges and some within cortex gray matter. There were caveats, however, when clusters of diffuse plaques having staining halos were misclassified as CAAs (Fig. 5b, CAA column). This was not entirely surprising as the project focused on cored plaques, so the CAA dataset was comparatively small; a larger CAA dataset containing the full spectrum of its morphologies may be a useful subject of further projects. Indeed, CAAs can be delineated into various staging schemes, such as by their location within the media of the vessel and vessel integrity[11], which is important in diagnosing CAA-related hemorrhage.

Using Phase III's larger field-of-view (3840 microns) hold-out regions, the overlays of CNN prediction confidence maps onto ground-truth annotations highlighted cases and context of prediction success and disagreement (Fig. 6b, Supplementary Fig. 9). Nonetheless, accurate predictions alone do not guarantee meaningful learning or that the model will be applicable to new scenarios or populations[60]. Plaque morphology can differ by neuroanatomic location—a CNN model developed from temporal gyri plaques may not be translatable to plaques in other anatomic areas, such as the striatum[1,61]. Although an explicit evaluation of all confounders is outside the scope of this work, the feature saliency and occlusion map studies (Fig. 7) demonstrated that the models focus on image features relevant for neuropathology. Guided Grad-CAM techniques near-exclusively highlighted the IHC stained regions, in patterns characteristic of the pathologies (Fig. 7, white-on-black maps). Complementarily, feature occlusion studies illustrated that the central amyloid core is the most discerning feature of a cored plaque's correct identification, and that its occlusion transforms a CNN model's classification to diffuse plaque. Importantly, the crucial features emerging from these machine learning introspection techniques —dense compact Aβ centers for cored plaques, ill-defined amorphous Aβ deposits for diffuse plaques, and Aβ within the media of the cortical vessels for CAAs—all agree with key features used by experts[1,11,62,63].

We finally evaluated whether CNN models could automatically quantify Aβ burden on a whole-slide level in a way that would correlate with standard semi-quantitative methods for plaque assessment (i.e., CERAD neuritic plaque scores). As true neuritic plaques are not distinguishable using Aβ-selective IHC stains, we leveraged CERAD-like manual scores (none, sparse, moderate and frequent) specific to each amyloid class. We found that a preliminary WSI-level CNN-based score we developed (Methods)

correlated strongly with manual CERAD-like scores (Fig. 8). CNN-based scores from one CERAD-like category were significantly different from WSIs in other categories (for cored plaque, $p < 0.01$, using two-sided Student's $t$-tests). Beyond its overall correspondence with CERAD, the finer-grained CNN-based metric captured subtle variations of Aβ burden within each CERAD category. The more detailed and sensitive measurement of Aβ burden, after appropriate validation in further studies, may strengthen statistical power for clinicopathological correlations[64]. Automated scores of this nature might be applied across entire archives of stained tissue from diverse anatomic regions, or aid in studies focused on evaluating burden specific to certain neuroanatomic locales or other local landmarks.

Several caveats, however, merit mention. Foremost among them is the intentional restriction of this proof-of-concept study's scope to annotations made by a single expert neuropathologist on a single immunohistochemical stain within a single anatomic region. Differences in experience and annotation criteria will likely result in individual expert variation among ground truth labels. The goal and intent of this project were therefore to establish the potential to extend an individual neuropathologist's plaque-identification capabilities in the context of their normal workflow. Furthermore, all data used in this study were from a single brain bank and retrieved and digitized under the same conditions; more diverse datasets from multiple sources will yield more robust and reliable models. We noted also that when the same hold-out set (Phase III) was annotated by web platform (Fig. 2c) versus entirely by hand, this resulted in at times differing labels (Supplementary Fig. 11). Future work may build on these foundations to investigate cross-neuropathologist plaque labeling, differing stains, anatomic regions, or collection centers, as well as region-level scoring systems to quantify bulk Aβ pathology burdens.

Taken together, the present study demonstrates a deep learning approach that can augment the expertize and analysis of an expert neuropathologist. Approximately 30 h of expert labeling yield a highly scalable and reusable CNN model capable of novel inference on unseen WSIs—going forward, 15 h may be sufficient to create new visualizations and understandings of Aβ pathology distribution. CNNs automatically learn relevant features from immunohistochemical image data and to exploit Alzheimer's disease pathological features in agreement with human-interpretable neuropathology. Significantly, CNN-based Aβ pathology burden agrees with CERAD-like scoring. Many brain banks have archival materials of stained slides arising from neuropathological diagnosis; models such as these may capitalize on such materials to help quantify pathologies providing pathological deep phenotyping in a scalable way. We anticipate collecting annotated datasets from multiple sources and experts will improve recall, sensitivity, and accuracy of the resulting neural network models and support training of more sophisticated model architectures. We hope this proof of concept motivates further work in this field, where automated pathology classification could have far-reaching impact; to this end, we make the CNN model code and dataset openly available to the community (see Data Availability).

## Methods

**Ethics approval and consent to participate**. These studies utilized only human post-mortem tissues. Only living subjects are defined as Human Subjects under federal law (45 CFR 46, Protection of Human Subjects). All participants or legal representative signed informed consent during the life of the participant as part of the University of California Davis Alzheimer's Disease Center program. All human subject involvement was overseen and approved by the Institutional Review Board (IRB) at the University of California, Davis. All data followed current laws, regulations, and IRB guidelines (such as sharing de-identified data that does not contain information used to establish the identity of individual deceased subjects).

De-identified data do not contain personal health information (PHI) like names, social security numbers, addresses, and phone numbers. Data were shared with a randomly generated pseudo-identification number.

**Case cohort**. All samples were retrieved from archives of the University of California, Davis Alzheimer's Disease Center (UCD-ADC) Brain Bank. Archival samples analyzed in this study were 5 μm formalin fixed, paraffin embedded, sections of the superior and middle temporal gyrus. The tissue had been previously stained with an Aβ antibody (4G8, recognizing residues 17–24, dilution 1:1600, BioLegend (formally Covance), catalog number SIG-39200) that were first pre-treated with formic acid to rid samples of endogenous protein. All slides were digitized using an Aperio AT2 up to ×40 magnification. Supplementary Table 1 details overall NIA Reagan criteria[65], CAA type[66] 1 or 2 within the section, and Thal Amyloid phase[5] of these cases.

Procedures were in accordance with ethical standards of the Helsinki Declaration. Operations of the University of California Davis Alzheimer's Disease Center was approved by the Institutional Review Board (IRB) of the University of California Davis, and written consent for autopsy was obtained for each participant during life. Details of this program have been previously published[67].

**Dataset splitting**. A total of 33 WSIs corresponding to 33 separate decedent cases, spanning all clinicopathologically-assigned NIA Reagan criteria, and possessing a variety of CERAD scores (see Supplementary Table 1) were used for model development and training (29 training, 4 validation images). An additional 10 WSIs were selected by an expert neuropathologist (BD) as a held-out test set and were not released until the model development phase of the study had been completed. Finally, a further 20 blinded WSIs were collected solely for use in the CERAD-like scoring comparison study and combined with the 10 test WSIs for the 30-WSI analysis reported in Fig. 8b. CERAD-like scores were available for all but one of the 63 WSIs used in this study; thus Fig. 8a uses 62 WSIs as a complete dataset.

**Image preprocessing**. All initial image preprocessing was performed in the open-source library PyVips[68]. Images were loaded at ×20 magnification, corresponding to a resolution of 0.5 microns per pixel (MPP). Slide color normalization was performed by the method of Reinhard et al.[69] using a reference image selected by the annotating expert (see Supplementary Fig. 1). The resulting WSIs were regularly tiled to 1536 × 1536 pixel tiles, corresponding to 768 × 768 micron regions of tissue for further analysis, resulting in a total of 33,111 tiles for the training set.

**Image segmentation**. Image segmentation was performed using the open-source library OpenCV[70]. Immunohistochemically-stained entities including cored plaques, diffuse plaques, and CAA appeared in the brown hue-region and segmentation was performed in the HSV colorspace utilizing a permissive colormask. For expert annotation, intracellular amyloid precursor protein (as denoted by cytoplasmic staining) was considered negative. Morphological opening and closing operations were performed to smooth the binary masks, and a standard blob-detection procedure was applied to isolate candidate objects. These unique components were center-cropped to a fixed size (256 × 256 pixels), corresponding to a region of 128 × 128 microns. This procedure resulted in nearly 500,000 images. Noisy background deposits were eliminated through a minimum stained area threshold of 1500 pixels (375 square microns), resulting in a total of 206,888 tiles.

**Plaque-labeling web interface**. To allow for the rapid and efficient annotation of the dataset, we developed a custom Python Flask web application that we deployed on Amazon Web Services Elastic Beanstalk[39]. The web-based interface allowed for remote login by the expert labeler, and enables fast, multi-label annotation of images using individual keystrokes. In the interface, images corresponding to 128 × 128 microns regions were shown to the annotator. A bounding box in the image specified which specific candidate object was to be labeled. Several elements of gamification, such as leveling, achievement badges (crown icon), and progress bar filling (green bar) were incorporated to motivate and track annotation task progress. A timestamp function was implemented to record the number of images per hour annotated by the expert (BD). All labels were stored in a SQL database using the Amazon Relational Database Service.

All images were annotated by a single neuropathologist (BD) and labeling of the image data proceeded in three phases: (1) In an initial phase, 55,000 images stemming from 3811 unique tiles were labeled; (2) In the second phase, images containing the minor classes of interest (cored plaques and CAAs) were enriched by running the CNN model built from the first-phase dataset on the remaining 101,671 images. These images were ranked by their predicted likelihood of containing cored plaques or CAA. We then chose the top 11,029 images for labeling. The labeled data from Phase I and Phase II were combined as the entire dataset (Phase I + II) for model training and evaluation. (3) In the third phase, two test sets were constructed with the same data but two distinct labeling methods. A 7680 × 7680 pixel (0.5 MPP) region was selected within each of the 10 hold-out test set WSIs by an expert neuropathologist as the area of interest. For the first test set, 10,873 candidate object tiles extracted from these 10 regions were labeled using the image-labeling web interface. For the second test set, the cored plaques and CAA

were directly marked by a neuropathologist on the selected region at a standard 10× (768 microns) visual field.

**Model development and training**. All neural network models were trained in the open-source package PyTorch[71] on four NVIDIA GTX 1080 or Titan X graphics processing units. Our optimized model used a simple convolutional architecture for image classification, consisting of alternating (3 × 3) kernels of stride 1 and padding 1 followed by max pooling (Fig. 3a), followed by two fully connected hidden layers (512 and 100 neurons) and rectified linear units as the nonlinear activation function. All neural network models were trained using backpropagation. The optimized training procedure used the Adam[72] optimizer with a multi-label soft margin loss function with weight decay (L2 penalty, 0.008) and dropout (probability 0.5 for the first two fully connected layers and probability 0.2 for all convolutional layers). Training proceeded with mini-batches of 64 images with real-time data augmentation including random flips, rotations, zoom, shear, and color jitter. When calculating the classification accuracy, a threshold of 0.91, 0.1, and 0.85 was used for cored plaque, diffuse plaque, and CAA prediction, respectively. Predictions with confidence above the threshold were considered to be positives.

**Prediction confidence heatmaps**. A sliding window approach[43] was applied with the trained CNN model to generate confidence heatmaps. At each step, the CNN model took a 256 × 256 pixel region as input and generated a prediction score for cored plaques, diffuse plaques, and CAAs. By systematically sliding the input region across the entire image, the prediction scores were plotted as prediction confidence heatmaps. The color represented the CNN's prediction confidence for the presence of cored plaques, diffuse plaques, and CAAs in the corresponding region, with yellow being the most confidence, and purple the least. We used a stride of 16, 4, and 1 for Fig. 5a–c, respectively.

**Guided gradient-weight class activation maps**. Guided gradient-weighted class activation mapping (Guided Grad-CAM)[47] was performed in PyTorch to generate saliency maps that highlight relevant features on the class of interest. The saliency map is a pointwise multiplication of guided backpropagation and Grad-CAM. Guided backpropagation produces a pixel-space gradient map of predicted class scores with respect to pixel intensities of the input image. Guided Grad-CAM produces a more class specific map which is the dot product of the feature map of the last convolutional layer and the partial derivatives of predicted class scores with respect to the neurons in the last convolutional layer. We employed an open-source implementation of Guided Grad-CAM by Ozbulak[73] along with our trained models to evaluate the learned features of the model on individual class examples (Fig. 7).

**Feature occlusion studies**. Feature occlusion studies[48] were performed to show the influence of occluding regions of the input image to the confidence score predicted by the CNN model. The occlusion map was computed by replacing a 16 × 16 pixels region of the image with a pure white patch and generating a prediction on the occluded image. As systematically sliding the white patch across the whole image (stride = 1 pixel), the prediction score on the occluded image was recorded as an individual pixel of the corresponding occlusion map. The color represented the CNN's prediction confidence for the presence of cored plaques, diffuse plaques, and CAAs of the occluded image, with red being the most confidence, and blue the least.

**Segmentation on prediction heatmaps and CNN-based scoring**. Prediction confidence heatmaps were segmented using the open-source library OpenCV[70]. First, a CNN confidence threshold was applied to the heatmaps, with only prediction confidences higher than the threshold retained, indicating the positive predictions of plaques. Morphological opening and closing operations were then performed to smooth the binary masks, and prediction areas exceeding a second threshold set to eliminate CNN-noise. Application of a standard blob-detection algorithm predicted discrete counts of the Aβ pathologies by the CNN model, which were then normalized by tissue area to provide CERAD-like scores. These hyperparameters, CNN confidence threshold (cored plaque: 0.1; diffuse plaque: 0.95; CAA: 0.9) and size threshold (cored plaque: 100; diffuse plaque: 1; CAA: 200 pixels), were optimized by statistical analysis on the training and validation sets.

**Tissue segmentation**. Tissue areas from WSIs were calculated using the open-source libraries PyVips and OpenCV. Tissue segmentation against the slide background was performed by applying a color mask in the lightness-chroma-hue (LCH) colorspace. Morphological opening and closing operations were performed to smooth the binary mask, and the tissue areas were calculated as the pixel sum of the refined mask.

**Statistical analyses**. Statistical analyses were performed using the open-source library SciPy (http://www.scipy.org/). Spearman rank-order correlation coefficient was calculated between CNN-derived scores and CERAD categories for the superior and middle temporal gyrus. A two-sided Student's t-test was used to test the null hypothesis that two independent samples have identical expected values.

CNN scores of WSIs from different CERAD categories were used for the test. Data were presented as box plots overlaid with individual data points. Box plots showed interquartile range (top and bottom of the box), median (central band), and outliers (points beyond the whiskers). Individual data points were shown as specific points. $p \geq 0.05$ was considered not significant (ns); *$p < 0.05$, **$p < 0.01$, ***$p < 0.001$.

**Reporting summary**. Further information on research design is available in the Nature Research Reporting Summary linked to this article.

### Data availability

We have made the full raw WSI dataset and the annotated plaque-level dataset (https://doi.org/10.5281/zenodo.1470797) openly available. A landing page is available at https://www.keiserlab.org/resources. The source data underlying Figs. 4a–d and 8a–d and Supplementary Figs. 6a–d and 7a, b are provided as a Source Data file.

### Code availability

Source code for the CNN model is available at https://github.com/keiserlab/plaquebox-paper/.

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

## Acknowledgements

The authors thank the families and participants of the University of California Davis Alzheimer's Disease Center for their generous donations. The authors would also like to thank the Department of Pathology and Laboratory Medicine at the University of California Davis for the use of their Aperio Scan scope and Kelsey Erickson and Justin Athey for aiding with slide scanning. This study was funded by a NIH P30 AG010129 grant (B.N.D., C.D., L.-W.J., and L.B.), a Paul G. Allen Family Foundation Distinguished Investigator Award (M.J.K.), and the China Scholarship Council (Z.T.). These agencies had no role in any aspect of the study, including study design, data collection, analysis, or writing.

## Author contributions

The studies were conceptualized, results analyzed, and manuscript drafted by Z.T., K.V. C., M.J.K., and B.N.D. Z.T. and K.V.C. wrote the code, performed the computational experiments, trained the models, and wrote the first draft of the manuscript, with guidance from M.J.K. and B.N.D. C.D., L.-W.J., and B.N.D. provided clinical and neuro-pathology work-up for case classifications and data interpretation. L.B. provided supervision of statistical analysis and interpretation of data. B.N.D. provided case selection and annotations, and neuropathology data interpretation. B.N.D. and M.J.K. provided overall study supervision. All authors were involved in critical revisions of the manuscript, and have read and approved the final version.
