## [Peer Review File · Nature Communications]

Reviewers' comments:

Reviewer #1 (Remarks to the Author):

I believe this is an important and interesting application of deep neural networks.

1- The paper is very hard to read and it was not easy to understand the steps. I think it would be better to transfer some methods to the supplement part to avoid confusion.

2- The code is publicly available and I tried to run the code. However, there are too many files and it is not easy to find them. I think it is necessary to have a readme that explains all steps. Better software management is needed before publication.

3- Based on the manuscript context, the majority of candidate images were annotated as diffuse plaque morphologies (84.8% of the annotations), with cored plaques (2.2%) and CAAs (1.1%) making up the minor classes. This means that a majority classifier that classifies all images with the label "diffuse plaque morphologies" would have an accuracy of approximately 85% and with a high AUC. For this reason, I suggest, to make a fair comparison, report the accuracy for simple baselines.

4- Regarding the low number of images with cored plaques and CAAs labels, I wondered if the authors tried to use different methods such as image rotation for increasing the number of images for the minor classes and compare the result with the method they used in phase II of their pipeline.

5- The performance decrease in the test set as compared to the validation set. It would be good to explain why assessing the trained algorithm with the validation set has higher performance compared to the test set. Also, there are significant differences between AUROC and AUPRC in the test set. My suggestion is to explain why AUROC is almost with no error while the AUPRC is 0.74. Maybe for classification projects, the accuracy is a better measure.

6- The author nicely applied a sliding window approach to generate WSI heatmaps of predictions to show the neuroanatomic location of pathologies. However, I wondered if they trained DNN using different image magnification (i.e. 20X vs. 10X vs. 5X) to compare the performance results. Previous research showed using different image magnification can affect the performance (PMID: 30224757; DOI: 10.1038/s41591-018-0177-5).

7- Are the image preprocessing steps important? How these preprocessing steps affect the final result? There must be a thorough discussion on this. I wondered if the authors tried the same steps without any preprocessing. Previous works have shown the preprocessing does not necessarily increase the accuracy (MCID: PMC5828543; DOI: 10.1016/j.ebiom.2017.12.026).

8- The authors have used a DNN with 6 layers. Would not be better to use deeper networks for classification?

9- Model Performance Improves Nonlinearly with the Number of Training Examples: The non-linearity claim is kind of non-rigorous statement. In my opinion, this is somewhat trivial because accuracy is always less than 100%. Therefore, it cannot increase linearly.

10- As expected, model performance positively tracked with the total number of training examples: This is a well-known fact in deep learning or generally in machine learning. I did not totally understand the point of this experiment though. Perhaps better to transfer this part to the supplementary.

11- The paper should be revised for some typo and grammatical errors. For example: "an dataset of approximately" should be "a dataset of approximately". Also, several a/the are missing in the text.

Reviewer #2 (Remarks to the Author):

The present manuscript focuses on the automated detection of different amyloid-beta plaques in human temporal cortex stain samples. There to, they applied a fully automated image segmentation approach and a novel deep learning model to identify plaques.

The manuscript is excellently written, sound, and contains various additional material and methodological explanations to substantiate the findings. Impressively, despite the vast amount of information and figures, the authors managed to keep a tight focus and at the same time illustrate nicely various major considerations that guided the analyses. The applied methods were fully adequate and state of the art with respect to deep learning and control for overfitting/overestimation of the classifier results. Nicely, additional saliency maps and occlusion analyses aid to explain the networks' sensitivity and driving image features. Notably, additional analyses on the effect of training samples and effect of rater experience nicely complement to the overall excellent quality of the present paper.

For the present state I only have minor issues that should be considered:

1. p.5 line 18: The two abbreviations IHC and WSI appear without being introduced before.
2. p.9 first paragraph, correlation with CERAD scores and Fig. 8: Would it be useful to additionally report a rank correlation coefficient for CNN-based scores and CERAD categories?
Fig. 8: Could you add a jitter to the sample dots to better visualize the distribution of values?
3. Fig. 3: What size is the convolutional filter size? For all layers 3x3 pixels? What is the configuration of the Maximum Pooling layers? (2x2 grid without stride?)
4. Fig. 7: Please add to the color scale in the bottom right corner that it presents prediction confidence.
5. Supplementary Table 1+2: What is the unit of the post mortem interval (PMI)? Do the plaque numbers (right columns) represent the CERAD scoring?
6. Supplementary Fig. 8+10: Please use a different figure legend/title, e.g. "Supplementary Figure 8 continued" for the second/third pages of the multipage figures.

Reviewer #3 (Remarks to the Author):

The authors of the study have developed and validated a convolutional neural network model that allows to distinguish between different types of amyloid pathology in a cohort of patients with and without Alzheimer's disease (AD). This is a very interesting study and absolutely required, as it provides original data to aid the further classification of AD cases based upon the morphological characteristics of pathological lesions present. The novel techniques may inform researchers of discrete clinico-pathological phenotypes not previously described, and may open up new avenues for therapeutic development. I have a few queries/minor comments the authors should consider: -

1. Considering this is a neuropathological study focussing on amyloid plaques, if available please could the authors present Thal phases in the demographics data in supplementary table 1.

2. Please can the authors clarify in the demographics table in supplementary table 1, if the presence of AD is based on a clinico-pathological or purely neuropathological diagnosis. Related to my point above, without specified Thal phases for the cases, case numbers 8 and 16 in supplementary data could not be classified as anything higher than low AD neuropathologic change, which may have consequences if this data were used in downstream analyses, please can this data be checked?

3. The antibody used to stain the tissue sections (4G8) is very well used in diagnostic and research laboratories, however even with pre-treatment of formic acid it will still stain physiological intracellular APP. Please can the authors clarify if any precautions were taken to distinguish between APP and smaller diffuse plaques?

4. CAA was also assessed within the present study, although as far as I can see the authors did not state whether type 1 CAA (inclusive of capillary CAA) was assessed. As capillary CAA has been shown to correlate well with AD neuropathologic criteria (i.e. Braak stages and CERAD scores) I think this would be a useful addition to the manuscript. At the very least the authors should clarify in the text what type of CAA was assessed in the analysis.

Response to Reviewer Comments

Reviewer 1

I believe this is an important and interesting application of deep neural networks.

We thank the reviewer for their support of this study's impact.

1.1 The paper is very hard to read and it was not easy to understand the steps. I think it would be better to transfer some methods to the supplement part to avoid confusion.

We thank the reviewer for the attention they have lavished on the manuscript and regret that some methods details may have detracted from the core narrative in their reading. This led to substantial discussion within our groups and to careful re-readings of the text. We made several edits for clarity. Ultimately, however, we were unable to identify large regions that should be excised, and as other reviewers praised the manuscript as "a very interesting study" (Rev 3) that was "excellently written" (Rev 2) with "a tight focus and at the same time illustrat[ing] nicely various major considerations" (Rev 2), we elected to avoid major reorganizations that could disrupt this at times difficult balance.

1.2 The code is publicly available and I tried to run the code. However, there are too many files and it is not easy to find them. I think it is necessary to have a readme that explains all steps. Better software management is needed before publication.

We appreciate the hands-on feedback. Correspondingly, we have created a more detailed README to better organize the code and to orient users of the referenced open source GitHub repository (<https://github.com/keiserlab/plaquebox-paper>). The revised README further clarifies the purpose and instructions for use (see the new "Demo" subsection) of each Jupyter notebook file, which themselves are in-depth commented and explained in-line.

For added user convenience we are also working on centralizing all of the Jupyter notebooks' data file-paths to point to a single new "data/" subfolder stub within the repository, wherein a user's separately-downloaded Zenodo data repository files may all be placed, although this requires regenerating all notebook calculations and their embedded figures. Thus this final repository accessibility enhancement will take place approximately one week past this revision's submission.

Our goal with this repository was to freely provide all demo code necessary for reproducible science, including all raw and processed neuropathological image data in the corresponding Zenodo repository (<https://doi.org/10.5281/zenodo.1470796>). This includes final trained model weights and annotated images, such that a PyTorch programmer can reproduce the paper's scientific results from the demo code and the raw input dataset. The intent of the current study, however, is neither to create nor to maintain a full-fledged turn-key software package. We regret that our own resource and time limitations preclude us from providing a resource more typical of large explicitly-funded or commercial projects.

1.3 Based on the manuscript context, the majority of candidate images were annotated as diffuse plaque morphologies (84.8% of the annotations), with cored plaques (2.2%) and CAAs (1.1%) making up the minor classes. This means that a majority classifier that classifies all images with the label "diffuse plaque morphologies" would have an accuracy of approximately 85% and with a high AUC. For this reason, I suggest, to make a fair comparison, report the accuracy for simple baselines.

The reviewer brings up an interesting point, but one that would result in devastatingly poor precision-recall performance (particularly in the recall of minor classes), and this is indeed the purpose of the precision-recall curves reported in Figure 4. Nonetheless, in the interest of comprehensiveness

the revised manuscript includes accuracy values for each plaque class in a new Supplementary Table 3 for readers interested in comparing metrics and summary metrics in the main text. E.g.,

“The overall classification accuracy was 0.970 on the validation set and 0.987 on the hold-out test set (Supplementary Table 3).”

More broadly, we agree with the reviewer that class imbalance can lead to misleading AUROC results, and this is too often overlooked. Our focus on reporting the corresponding AUPRCs for each result stems directly from this shared concern. Saito and Rehmsmeier (PloS One, 2015, doi: 10.1371/journal.pone.0118432) for instance discuss AUPRC’s benefits over AUROC and accuracy for unbalanced datasets. Furthermore one aspect of AUPRC and AUROC that we tend to prefer over single-point metrics such as accuracy is that the latter is a function of a single choice of confidence threshold, whereas the former two approaches plot performance tradeoffs of the model across all such thresholds. We thank the reviewer for bringing up these subtle but critical points.

1.4 Regarding the low number of images with cored plaques and CAAs labels, I wondered if the authors tried to use different methods such as image rotation for increasing the number of images for the minor classes and compare the result with the method they used in phase II of their pipeline.

Great point. As detailed in the Methods, we used standard PyTorch data augmentation techniques inclusive of image rotation, with precisely the suggested minority class upsampling approach, to achieve the performance reported in this study. Supplementary Figure 5 illustrates the data augmentation and upsampling procedure.

1.5 The performance decrease in the test set as compared to the validation set. It would be good to explain why assessing the trained algorithm with the validation set has higher performance compared to the test set. Also, there are significant differences between AUROC and AUPRC in the test set. My suggestion is to explain why AUROC is almost with no error while the AUPRC is 0.74. Maybe for classification projects, the accuracy is a better measure.

*Modern convolutional neural networks require significant hyperparameter optimization, which are tuned with the researcher being aware of performance on the validation set. Furthermore, the validation set is often a random subset of the same parent dataset from which the training set was derived. Due to these considerations, and because validation set performance is explicitly used to guide model selection, validation error is almost always lower than that of the stricter held-out test set, which by contrast provides an unbiased estimate of generalization error (per Hastie, Tibshirani and Friedman, *The Elements of Statistical Learning*, p. 222).*

As noted in our response to 1.3 above, accuracy and AUROC metrics have been shown to be poor indicators of model performance on imbalanced datasets (Saito and Rehmsmeier, PloS One, 2015, doi: 10.1371/journal.pone.0118432). AUROC summarizes the trade-off between true positive rate and false positive rate, which in an imbalanced dataset such as the current study’s is always high due to the high prevalence of diffuse plaques. AUPRC, on the other hand, plots the trade-off between precision and recall, which focuses entirely on positive class predictions, and thus better represents the actionability and class-imbalance corrected performance of the classifiers. Consequently, we expect AUPRC and AUROC to differ in the observed way, as AUPRC is the more difficult metric for this dataset. Parenthetically, we were pleasantly surprised by the high AUPRC performance of 0.74, as compared to other studies in the literature that do report AUPRCs.

1.6 The author nicely applied a sliding window approach to generate WSI heatmaps of predictions to show the neuroanatomic location of pathologies. However, I wondered if they trained DNN using different image magnification (i.e. 20X vs. 10X vs. 5X) to compare the performance results. Previous research showed using different image magnification can affect the performance (PMID: 30224757; DOI: 10.1038/s41591-018-0177-5).

We agree with the reviewer (and with the cited work) that substantial changes to image magnification should have a marked impact on prediction performance. A thorough study of this effect, however, would need to take into account the consideration that lower magnifications would by necessity not only contain fewer visual features (pixels) per amyloid object, but also result in fewer training tiles of unique objects across the entire dataset. Furthermore, it would likely be important to consider the effect of magnification on neuropathologist annotation, perhaps by intra- or inter-rater reliability studies. This is an interesting research direction to investigate in the future, but one that falls outside of the scope of the current study's experimental design.

1.7 Are the image preprocessing steps important? How these preprocessing steps affect the final result? There must be a thorough discussion on this. I wondered if the authors tried the same steps without any preprocessing. Previous works have shown the preprocessing does not necessarily increase the accuracy (MCID: PMC5828543; DOI: 10.1016/j.ebiom.2017.12.026).

We thank the reviewer for directing us to this reference. Accordingly, we have performed the suggested experiment, by retraining our models from the ground up without the color normalization step and evaluating their comparative performance, reported in the new Supplementary Figure 7. We found that color normalization had no significant impact on model performance in our study (e.g., compare Figure 4b vs. Supplementary Figure 7d).

Having conducted further literature search on this intriguing topic, we find that deep learning for histopathological analysis is an evolving field, with little current literature consensus on the importance of stain normalization as a preprocessing step for convolutional neural networks. For an example that, by contrast, recommends stain normalization, see: Ciompi et. al, <https://arxiv.org/pdf/1702.05931.pdf>.

However, in our study, color normalization serves a key additional purpose: namely, to reduce variation across whole slide images prior to manual annotation. The image preprocessing step was deemed helpful by our expert annotator, and in this proof of concept work, we chose to include it. A detailed follow-on study focusing on the question of color normalization might separately be designed to test annotation consistency as a function of normalization.

1.8 The authors have used a DNN with 6 layers. Would not be better to use deeper networks for classification?

The observation that deeper neural networks perform well typically comes from larger, more diverse datasets such as the ImageNet Object Recognition challenge, where there are over 1000 categories and millions of training examples (e.g., see Simonyan and Zisserman, ref 54 in the manuscript). Deeper networks operate, however, across more parameters and risk overfitting. With six layers, we were already able to overfit the models after a surfeit of training epochs, and consequently were careful to apply standard early-stopping convergence criteria and weight regularization techniques including dropout to improve generalization. Deeper network architectures that we investigated (not reported) overfit more easily and did not lead to better performance in our hands. Consequently we prefer to operate on a principle of parsimony, although it is certainly not impossible that a deeper architecture might nonetheless achieve yet higher performance. To this end, we provide the full annotated dataset to the community at <https://doi.org/10.5281/zenodo.1470796>, if others wish to try their hand at it.

1.9 Model Performance Improves Nonlinearly with the Number of Training Examples: The non-linearity claim is kind of non-rigorous statement. In my opinion, this is somewhat trivial because accuracy is always less than 100%. Therefore, it cannot increase linearly.

We agree that a property of the curve is eventual performance saturation. However, the key point of this experiment was simply that model performance reached saturation after a quantifiable number of training examples (e.g. diminishing returns from increased annotations). This result is actionable, and contradicts the less specific "more is always better" preference so commonly encountered in other

application domains of machine learning, especially when there is a high opportunity cost (such as an expert neuropathologist's limited labeling time).

1.10. As expected, model performance positively tracked with the total number of training examples: This is a well-known fact in deep learning or generally in machine learning. I did not totally understand the point of this experiment though. Perhaps better to transfer this part to the supplementary.

We agree that it is a well-known expectation that machine- and deep learning model performance should increase as a function of the number of pertinent training examples. However, we found few studies carefully quantifying this effect in general, and fewer still in this application domain, to our knowledge. Consequently, we contend that these curves importantly report on the precise tradeoff between the annotation time dedicated by an expert neuropathologist and the performance of the resulting models. Expert annotation is highly valuable (Discussion) and this study allowed us to identify the point of diminishing returns during the annotation process. Additionally, this study illustrated the effect of minor class enrichment on model performance.

Secondarily, in Figure 4c, these dataset-size sensitivity analysis experiments comprised 5 model replicates trained from the ground up per dataset-size point, for 18 such points, for a total of 90 independently trained randomly-seeded and randomly-dataset-subsetted models. We hope this inspires other studies that venture into new application domains to consider a similar level of empirical quantification and consider its demonstration an important contribution.

1.11 The paper should be revised for some typo and grammatical errors. For example: “an dataset of approximately” should be “a dataset of approximately”. Also, several a/the are missing in the text.

We thank the reviewer for pointing out these grammatical infelicities. We have corrected them and have again proofread the entire text.

Reviewer 2

The present manuscript focuses on the automated detection of different amyloid-beta plaques in human temporal cortex stain samples. Thereto, they applied a fully automated image segmentation approach and a novel deep learning model to identify plaques.

The manuscript is excellently written, sound, and contains various additional material and methodological explanations to substantiate the findings. Impressively, despite the vast amount of information and figures, the authors managed to keep a tight focus and at the same time illustrate nicely various major considerations that guided the analyses. The applied methods were fully adequate and state of the art with respect to deep learning and control for overfitting/overestimation of the classifier results. Nicely, additional saliency maps and occlusion analyses aid to explain the networks' sensitivity and driving image features. Notably, additional analyses on the effect of training samples and effect of rater experience nicely complement to the overall excellent quality of the present paper.

We thank the reviewer for their kind words and support for the work.

For the present state I only have minor issues that should be considered:

2.1 p.5 line 18: The two abbreviations IHC and WSI appear without being introduced before.

We thank the reviewer for their attention to clarity. We have included the full wording at the first appearance of these abbreviations: Immunohistochemistry (IHC) and whole slide images (WSI) within the revised text.

2.2 p.9 first paragraph, correlation with CERAD scores and Fig. 8: Would it be useful to additionally report a rank correlation coefficient for CNN-based scores and CERAD categories?

We agree and have incorporated this suggestion. The new Supplementary Table 4 reports the Spearman rank-order correlation coefficient based analysis for comparison.

2.3 Fig. 8: Could you add a jitter to the sample dots to better visualize the distribution of values?

Great point. We have added jitter and updated Figure 8 to better visualize the distribution of values.

2.4 Fig. 3: What size is the convolutional filter size? For all layers 3x3 pixels? What is the configuration of the Maximum Pooling layers? (2x2 grid without stride?)

Yes, the entire network used 3x3 convolutions and 2x2 max pooling (Methods). For clarity, the revised Figure 3 caption now reports the model architecture.

2.5 Fig. 7: Please add to the color scale in the bottom right corner that it presents prediction confidence.

As suggested, we have updated Figure 7's color scale label for clarity.

2.6 Supplementary Table 1+2: What is the unit of the post mortem interval (PMI)? Do the plaque numbers (right columns) represent the CERAD scoring?

We thank the reviewer for this comment. We have added to the table the description of PMI being in hours. The right columns (Cored Plaques MTG, Diffuse MTG and CAA MTG) do represent CERAD-like scores (using the denotation none=0, sparse=1, moderate=2, and frequent=3). We have adjusted the text for consistency and added these denotations to the tables.

2.7 Supplementary Fig. 8+10: Please use a different figure legend/title, e.g. "Supplementary Figure 8 continued" for the second/third pages of the multipage figures.

We have updated the Supplementary Figure legends and titles as requested in the revised Supporting Information.

Reviewer 3

The authors of the study have developed and validated a convolutional neural network model that allows to distinguish between different types of amyloid pathology in a cohort of patients with and without Alzheimer's disease (AD). This is a very interesting study and absolutely required, as it provides original data to aid the further classification of AD cases based upon the morphological characteristics of pathological lesions present. The novel techniques may inform researchers of discrete clinico-pathological phenotypes not previously described, and may open up new avenues for therapeutic development. I have a few queries/minor comments the authors should consider:

We thank the reviewer for their support of this study.

3.1 Considering this is a neuropathological study focussing on amyloid plaques, if available please could the authors present Thal phases in the demographics data in supplementary table 1.

We thank the reviewer for this comment. We have completed additional 4G8 staining on neuroanatomic regions and have included the Thal Phases for cases listed in Supplementary Table 1. As these were archival cases, materials were not available in all regions for determining the exact Thal Amyloid phase (Thal phase) on 2 cases, as marked in Supplementary Table 1.

3.2 Please can the authors clarify in the demographics table in supplementary table 1, if the presence of AD is based on a clinico-pathological or purely neuropathological diagnosis. Related to my point above, without specified Thal phases for the cases, case numbers 8 and 16 in supplementary data could not be classified as

anything higher than low AD neuropathologic change, which may have consequences if this data were used in downstream analyses, please can this data be checked?

We thank the reviewer for this very careful assessment and review. The purpose of this study was to use archival materials to develop a pipeline for image recognition of amyloid deposits (so as to provide a cost-effective way for brain banks to gain more quantitative data on materials already accumulated). A large majority of the cases (including cases #8 and 16) were neuropathologically assessed prior to many of the most recent diagnostic entities and criteria (such as NIA-AA, CART, PART, etc.). Many brain bank databases have developed in an organic matter, and there can be data entry without support for quality control mechanisms. Furthermore, another common issue with tissue banks is that diagnostic criteria change over time and it is typically not feasible to consistently update cases to conform to current diagnostic criteria. The authors appreciate the reviewer pointing out these discrepancies.

Case #8 was a Braak NFT stage III with a Thal phase of 5, overall CERAD of sparse, and categorized as NIA Reagan Low. The case was demented proximal to death. Other pathological changes included those warranting a diagnosis of Lewy body dementia (having moderate numbers of Lewy bodies within the substantia nigra and cortex).

Case #16 was a Braak NFT stage II, with overall CERAD moderate, Thal phase 3 and categorized as NIA Reagan of Low. The participant had questionable cognitive impairment proximal to death. Other denoted pathologies included a microhemorrhage at the pontine nuclei and an infarct in the reticular formation of the dorsal medulla. No Lewy bodies were present. Although considered Low with NIA Reagan, this case did not meet clinicopathologic criteria for any other dementia (at the time of diagnosis where many other diagnoses such as CART, PART, etc. were not revealed). Upon Clinical pathology conference the case was categorized as CERAD possible AD and vascular disease other.

We thank the reviewer immensely for the keen eye. We have re-evaluated all cases listed within Supplementary Table 1 to reconcile any diagnostic and categorical discrepancies. The presence of AD was based on a clinicopathological diagnosis. As we had complete data on Braak and CERAD scores for all but one case, we have changed the AD column in Supplementary Table 1 to denote a clinicopathological diagnosis using NIA Reagan criteria of the following categories: Not AD, Low, Intermediate, or High. We have also added these methods and proper citations to the revised manuscript.

3.3 The antibody used to stain the tissue sections (4G8) is very well used in diagnostic and research laboratories, however even with pre-treatment of formic acid it will still stain physiological intracellular APP. Please can the authors clarify if any precautions were taken to distinguish between APP and smaller diffuse plaques?

The authors thank the reviewer to provide further clarification. We have added the following to our methods:

“For expert annotation, intracellular amyloid precursor protein (as denoted by cytoplasmic staining) was considered negative.”

Furthermore, we have described in methods the following, to isolate candidate objects bounding boxes were center-cropped to a fixed size (256 x 256 pixels), which corresponded to a region of 128 x 128 microns. Noisy background deposits were also eliminated through a minimum stained area threshold of 1500 pixels (375 square microns).

3.4 CAA was also assessed within the present study, although as far as I can see the authors did not state whether type 1 CAA (inclusive of capillary CAA) was assessed. As capillary CAA has been shown to correlate well with AD neuropathologic criteria (i.e. Braak stages and CERAD scores) I think this would be a useful

addition to the manuscript. At the very least the authors should clarify in the text what type of CAA was assessed in the analysis.

We thank the reviewer for this point. The initial analysis of CAA was categorized solely as present or absent. We have added a column in Supplementary Table 1 denoting the CAA type (1 or 2 based on Thal DR et al. 2002 Journal of Neuropathology and Experimental Neurology) for the MTG section for each case denoted to have CAA by the expert neuropathologist. Of the annotated CAA images, 36.3% were capillary in origin and 63.7% were non-capillary. We have added these data to the revised manuscript.

With respect to CNN model development of CAA subtype, this is an excellent direction. We did not assess models that differentiated between these subtypes, as the present study focused on whether these major classes could be differentiated. We hope to address subtype classification in future work.

REVIEWERS' COMMENTS:

Reviewer #1 (Remarks to the Author):

The authors have addressed all my comments and the new version is much better compared to the former one. I have no more comments.

Reviewer #2 (Remarks to the Author):

The authors substantially revised and improved the manuscript and sufficiently addressed all of the reviewers comments raised for the previous version. From my point of view, the manuscript is excellently written, sound, concise, and can be published.

I only have a very minor point, that in Supplementary Table 3, the class confidence threshold for detecting diffuse plaques should be higher than the given 0.1?

Reviewer #3 (Remarks to the Author):

The aims of the study laid out in the manuscript were to create a technique to accurately evaluate amyloid deposits in human post-mortem tissue based on morphological appearance. The study is novel, reproducible, and in my opinion is of great interest to the field of neuroscience.

The authors have addressed my comments, have revised the manuscript accordingly, and I am therefore happy to recommend for publication.

Response to Reviewer Comments

Reviewer 1

1.1 The authors have addressed all my comments and the new version is much better compared to the former one. I have no more comments.

We thank the reviewer for their helpful suggestions and appreciate their review.

Reviewer 2

2.1 The authors substantially revised and improved the manuscript and sufficiently addressed all of the reviewers comments raised for the previous version. From my point of view, the manuscript is excellently written, sound, concise, and can be published.

We thank the reviewer for their support and endorsement of this study.

2.2 I only have a very minor point, that in Supplementary Table 3, the class confidence threshold for detecting diffuse plaques should be higher than the given 0.1?

We thank the reviewer for this point. The diffuse plaque confidence threshold of 0.1 was chosen by optimizing the F1 score, which is the harmonic mean of precision and recall, on the validation set. For completeness, we have exhaustively recomputed the analysis across a range of thresholds for all three plaque classes, reported in new Supplementary Tables 4 and 5. We find diffuse plaque accuracy varies by no more than 2-5% across a wide range of threshold choices. This is unsurprising as the dataset is imbalanced, with over 86% diffuse plaque and less than 5% cored plaque and CAA classes. Under this prior distribution, we would expect F1 score and accuracy to be less informative than multi-threshold measures such as AUROC and AUPRC.

We posit that the diffuse-plaque prediction benefited (slightly) from the more permissive confidence threshold because these plaques are not discrete and well-defined objects, unlike the other two classes. However, we have not endeavored to investigate this hypothesis in the current work, especially as the effect is small.

Reviewer 3

3.1 The aims of the study laid out in the manuscript were to create a technique to accurately evaluate amyloid deposits in human post-mortem tissue based on morphological appearance. The study is novel, reproducible, and in my opinion is of great interest to the field of neuroscience. The authors have addressed my comments, have revised the manuscript accordingly, and I am therefore happy to recommend for publication.

We thank the reviewer for their contributions and appreciate their endorsement of the study's novelty and significance.